# Multiple timescales of temporal context in risky choice: Behavioral identification and relationships to physiological arousal

**Hayley R. Brooks** [1,2], **Peter Sokol-Hessner** [1] *

1 Department of Psychology, University of Denver, Denver, Colorado, United States of America,
2 Department of Cognitive, Linguistic, and Psychological Sciences, Brown University, Providence, Rhode Island, United States of America

* Peter.Sokol-Hessner@du.edu

## Abstract

Context-dependence is fundamental to risky monetary decision-making. A growing body of evidence suggests that temporal context, or recent events, alters risk-taking at a minimum of three timescales: immediate (e.g. trial-by-trial), neighborhood (e.g. a group of consecutive trials), and global (e.g. task-level). To examine context effects, we created a novel monetary choice set with intentional temporal structure in which option values shifted between multiple levels of value magnitude ("contexts") several times over the course of the task. This structure allowed us to examine whether effects of each timescale were simultaneously present in risky choice behavior and the potential mechanistic role of arousal, an established correlate of risk-taking, in context-dependency. We found that risk-taking was sensitive to immediate, neighborhood, and global timescales: risk-taking decreased following large (vs. small) outcome amounts, increased following large positive (but not negative) shifts in context, and increased when cumulative earnings exceeded expectations. We quantified arousal with skin conductance responses, which were related to the global timescale, increasing with cumulative earnings, suggesting that physiological arousal captures a task-level assessment of performance. Our results both replicate and extend prior research by demonstrating that risky decision-making is consistently dynamic at multiple timescales and that the role of arousal in risk-taking extends to some, but not all timescales of context-dependence.

## Introduction

In risky monetary decision-making, values and probabilities associated with potential outcomes are critical to agents' choices, but recent evidence has established that temporal context, or recent events, also influence financial risk-taking. Though contextual sensitivity is common in affect [1, 2] and cognition [3, 4], its role in risk-taking is perplexing. When values and probabilities are independent and explicitly known, as in risky decision-making [5], any influence of recent events on subsequent choices would be disadvantageous. That contextual effects persist in risk-taking despite appearing disadvantageous suggests that risky decision-making may

**Data Availability Statement:** All data from this paper can be accessed at https://osf.io/a7nvx/. For all data cleaning and analysis scripts: https://github.com/sokolhessnerlab/vic.

**Funding:** The author(s) received no specific funding for this work.

**Competing interests:** The authors have declared that no competing interests exist.

be fundamentally contextually sensitive. Here we sought to characterize contextual effects in risk-taking and examine their possible relationship to affective mechanisms (physiological arousal).

The phrase "temporal context" is used here in a broad manner to refer to recent events. This usage is distinct from specific constructs in memory research [6] and meant to distinguish temporal from other kinds of context, for example, the current context (e.g. other options under consideration at a given moment).

Recent events appear to influence subsequent risky choices on (at least) three timescales: immediate (e.g. feedback from the previous trial), neighborhood (e.g. values from several consecutive recent trials), and global (e.g. cumulative earnings relative to evolving expectations).

Studies with immediate, trial-by-trial feedback have found that outcomes have a powerful but short-lasting effect on subsequent risky actions and the subjective value of risky options [7–14]. That these effects occur in highly-instructed, fully-informed settings that discourage the representation of temporal contexts suggests that they are part of the fundamental computations underlying risky decision-making and not demand effects or learning. However, understanding precisely how outcomes affect subsequent risk-taking is complicated by the fact that most studies with trial-by-trial feedback do not address these effects [15–19]. For more discussion of these effects see the Supporting Information.

Studies using blocks or sequences of similar choices to create contexts provide evidence for neighborhood-level context effects. In these studies, valuation [20] and risk-taking [21–23] were relative to the current neighborhood context (defined by e.g. a series of choice options with high vs low value) [23]. For example, individuals gave higher ratings for the same snack item following exposure to blocks of low-value snacks relative to blocks of high-value snack [20], an effect that was replicated with evaluations of risky gambles [21]. These studies demonstrated that when people evaluate choice options, they take into account recently-encountered values, approaching options that appear better relative to recent history, and vice versa. Despite the potential importance of such dynamic behavior in risky choice, only a handful of studies [23–25] to our knowledge examine how risky choices in the presence of feedback change as a function of the neighborhood context, let alone the extent to which these effects may differ from or be present alongside other timescales of temporal context.

Finally, the role of a dynamic global reference point, like cumulative earnings, has been long theorized [26], but rarely demonstrated. Most studies of risky decision-making actively design *against* global temporal context, for example by adopting payout structures other than cumulative earnings (like the outcome from a single trial or a subset of randomly-selected trials) [7, 23, 27–30]. Whether this payout structure actually prevents effects of cumulative earnings is complicated by the fact that, like immediate contextual effects, most studies do not analyze and report the influence of cumulative earnings on risky decision-making [7, 23, 27–30]. The few studies that have examined the presence of global context have robustly found that cumulative earnings are associated with changes in both laboratory [11, 31] and real-world [32, 33] risk-taking. That the directionality of the global context effect changes across studies, with higher cumulative earnings associated with both increased [32] and decreased risk-taking [11, 31, 33] suggests that cumulative earnings themselves may be only one component of global context–the other being the reference to which earnings are compared [34]. Despite the centrality of the reference point to some of the dominant modern theories of risky decision-making (like Prospect Theory (PT) [26], to our knowledge, only two studies have created and tracked dynamic reference points [35, 36] but neither examined the simultaneous presence of context effects at multiple timescales.

In the current study, we designed a novel choice set with intentional temporal structure to examine whether and how risky decision-making varies as a function of temporal context at

immediate, neighborhood, and global timescales, and address two major gaps in the literature. First, no studies have examined the simultaneous presence of all three timescales of contextual influence (or their possible interaction). Second, the underlying mechanisms supporting the effects of temporal context on risk-taking are as yet unknown, critical though they may be to resolving outstanding questions in the literature.

One particularly compelling candidate mechanism that we examine here is that of physiological arousal, known to be related to risky decision-making [37–39]. To measure physiological arousal, we use skin conductance responses (SCRs), an objectively quantifiable measure that reflects non-valenced sympathetic nervous system arousal responses [40]. The relationship between SCRs and risky monetary decision-making may be complex, with SCRs related to subjective value [27], levels of risk [41, 42], risk-taking [29, 43], reward anticipation [44], and loss relative to gain outcomes [29, 43, 45]. Because of this deep connection to risky choice, it is possible that SCRs also carry information about recent events occurring at one or multiple timescales, though this role has not been investigated.

The temporal characteristics of SCRs make them an additionally compelling candidate mechanism for temporal context. SCRs last several seconds, possibly long enough to span multiple trials. That the timescales of task events related to SCRs vary widely, from seconds [27, 29, 41–46] to several minutes [47] suggests that SCRs may relate to context occurring at multiple timescales in risky decision-making, possibly by carrying information or signaling changes in context.

In the current study, to examine the relationship(s) between physiological arousal and the three timescales of temporal context as outlined above, we additionally measured SCRs to decision and outcome events throughout our novel, temporally-structured risky decision-making task.

## Method

Below, we report how we determined the sample size, and all data exclusions, manipulations, and study measures. Data from this paper can be accessed at: https://osf.io/a7nvx/ [48]. For all data cleaning and analysis scripts: https://github.com/sokolhessnerlab/vic [49]. Data were analyzed using the package "lme4" [50] in R, version 4.0.0. [51]. The study's design and analysis were not pre-registered.

### Novel task and SCR

**Procedure.**    We collected SCRs for 62 participants during a risky gambling task (50 females, 1 trans male, and 11 males, mean age: 19.32(1.7), median age: 19, range: 18–29). Participants were recruited through the SONA system at the University of Denver between February 22, 2018 and November 5, 2018. Participants provided written informed consent prior to participation, and received course credit for the 1-hour session plus a monetary bonus from one randomly selected outcome from the task [52]. Participants completed a risky gambling task in a fully-instructed laboratory setting, producing informed, incentive-compatible, and robust repeated-measures individual-level choice data. After removing missed trials (a total of 120 trials were missed across 36 of the 62 participants with the median participant missing 1 trial), the dataset comprised a total of 14,004 choices. All procedures involving human participants were in accordance with the ethical standards of the institutional and/or national research committee and with the 1964 Helsinki Declaration and its later amendments or comparable ethical standards. The study was approved by the University of Denver's Institutional Review Board (No. 960405). Informed consent was obtained from all individual participants included in the study.

**Behavioral power analysis.** Our sample size was determined in part by (and in most cases exceeded) sample sizes of previous research in the domain of risk-taking [15, 27, 29, 53, 54] and the financial constraints associated with collecting incentive-compatible data. We used the "simr" package in R [55] a posteriori to confirm that our sample size was sufficient for detecting temporal context effects using generalized linear mixed effects models [50]. For example, for model 2 reported in the Results section, the power to detect the effect of previous outcome on risk-taking in our data using a linear mixed effect model was 88% (95% CI = [79.98% 93.64%], $\alpha = 0.05$).

**Task.** Our novel task consisted of 171–240 gain-only choices (most participants, N = 49, completed 240 trials; see the Supporting Information) between a risky gamble, with equal (p = 0.5) probabilities of a possible outcome of $0 and a possible positive outcome ranging from $.50 to $70, and a safe amount, received with p = 1 if selected, ranging from $.25 to $35. On each trial, the expected value (EV) of each option was the sum of each possible outcome's probability (i.e. p = .5 or p = 1) multiplied by the respective dollar amount. For example, for a trial with potential risky outcomes of +$30 and $0, and a safe amount of $16, the EV of the gamble is $15 (.5 x $30 + .5 x $0) and the EV of the safe amount is $16 (1 x $16). Choice options were displayed on the screen for 2 seconds followed by a response window of 2 seconds. There was a brief .5 second interstimulus interval (ISI) immediately following each response. Following the ISI, the outcome was displayed for 1 second, followed by a variable (1.5s-4.5s) intertrial interval (ITI) before the next trial began to reduce the strength of anticipation of the start of the next trial and to capture the delayed onset of SCRs [56]. Any remaining time from the 2 second response window (i.e. 2s –reaction time) was added to the ITI (Fig 1).

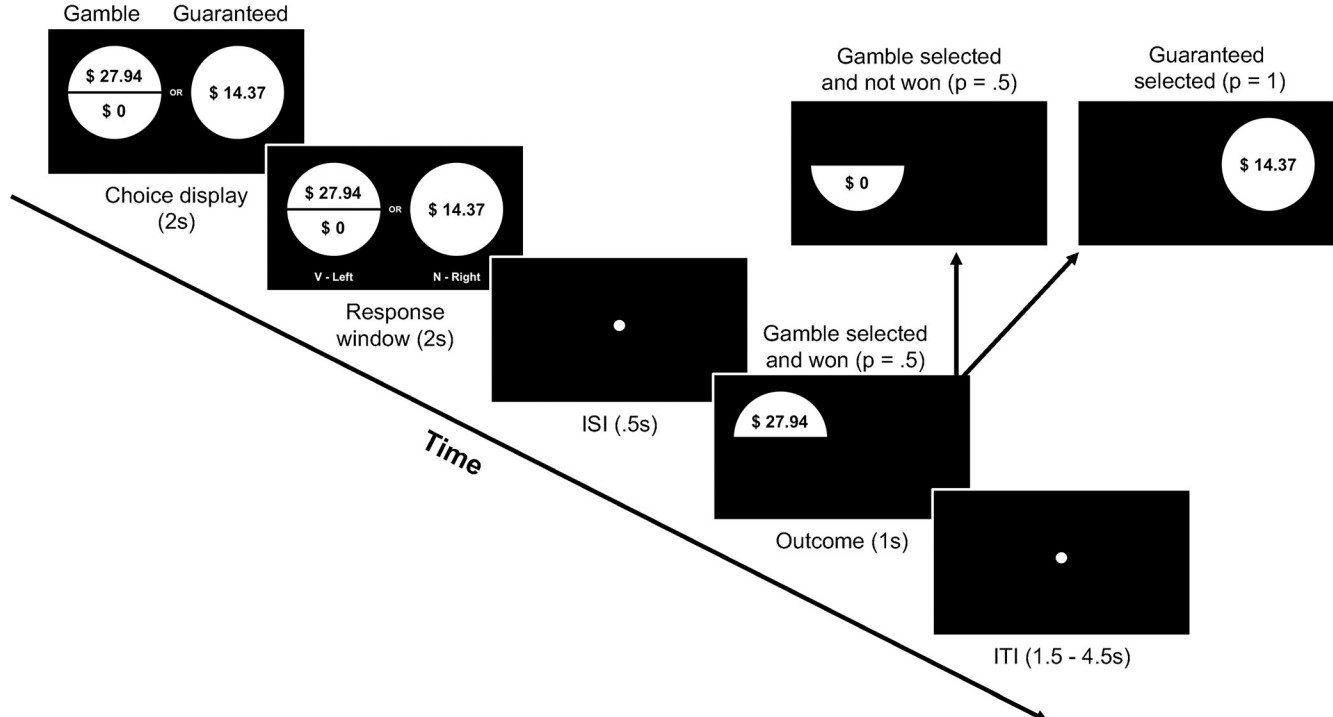

**Fig 1. Risky monetary decision-making task trial timing and events.** Participants made decisions between risky and safe options on each trial. The risky option consisted of two possible outcomes, each occurring with a 50% probability. The safe, or guaranteed option, featured a single outcome with 100% probability if selected.

The location of the risky gamble and safe options on the screen (left or right) were randomized across trials.

We designed the task with a unique temporal structure to create and measure temporal context at three timescales: immediate (outcome amounts on the previous trial), neighborhood (runs of trials with a shared expected value followed by shifts in value), and global (cumulative earnings relative to expectations). First, the immediate effects of temporal context were captured by the influence of each outcome on the subsequent choice, an effect identified in [8]. Because the task in the current study is nearly identical to that in [8], we expected to replicate the previously identified effect of past outcome on decision-making with risk-taking decreasing as previous outcome amounts increase. Second, we created context at a neighborhood timescale by organizing choices into runs within which the values of options were all near a common mean EV, or EV level. There were 9 to 10 runs, each varying in length from 9 to 36 trials long (all minor discrepancies across participants in task parameters are described in the Supporting Information). Following each run, there was a positive or negative shift to the next run's EV. Within each run, risky gain and guaranteed alternative values on each trial were derived from the EV level with some noise (up to +/- $2) which made risky gain values roughly double the guaranteed alternative values on each trial. For example, for a given run of 10 trials that shared an EV of $20, the guaranteed alternative values ranged from $18 to $22 (i.e. $20 +/- $2) and the risky gain values ranged to $36 and $44 (i.e. ($20 +/- $2) x 2). Possible shift values between runs ranged in magnitude (in EV) from +$5 to +$15 (positive shifts) and -$5 to -$15 (negative shifts) in increments of $1.25 (Supporting Information). For example, a shift of +$10 in EV increased the value of the guaranteed options by $10, and the risky options by $20. By creating multiple changes in context throughout the task, we were able to examine how risk-taking changed as a function of temporal context at an intermediate timescale, over multiple trials (Fig 2). Consistent with previous research that individuals tend to approach positive changes and avoid negative changes in valuation and decision-making tasks [20, 21], we expected risk-taking to increase following a positive shift (I.e. the expected value on the current trial is larger than that on the recent run) and decrease following a negative shift (i.e, the expected value on the current trial is smaller than that on the recent run). Lastly, the global timescale of temporal context, or cumulative earnings, was not inherently built into the task as cumulative earnings were not displayed to the participant at any time, and participants' behavioral bonus was determined by a single trial (and not their cumulative earnings). However, because the trial-by-trial feedback meant that participants could have latently tracked cumulative earnings (explicitly or implicitly) throughout the task, we examined how risk-taking changed as a function of cumulative earnings relative to expected earnings (which we assumed to be a linear term capturing increasing earnings over the task). Despite the latent nature of these variables, we expected individuals to track earnings [11] relative to expectations but the direction of this relationship is unclear. Previous research demonstrates when earnings increase above a certain threshold (e.g. expectation or reference point), risk-taking increases (e.g. "house-money effect") [12–14], and when earnings are less than a threshold, risk-taking decreases ("play it safe effect") [13]. See Discussion for additional possibilities [12–14]. We therefore hypothesized that when earnings were more than expected, individuals would take more risks and when earnings were less than expected, individuals would take fewer risks. The gambling task was displayed in MATLAB 2018b using the Psychophysics Toolbox [57].

**Physiological arousal.** For analyses of SCRs following outcomes, 15 participants were excluded for failing to show consistent SCRs following more than 25% of outcomes [27, 58–60] and one other participant was excluded for equipment malfunction for a remaining total of 46 participants for SCR analyses. Roughly half of the 10,304 trials for the 46 remaining participants (classified as "responders") had skin conductance responses following outcomes that

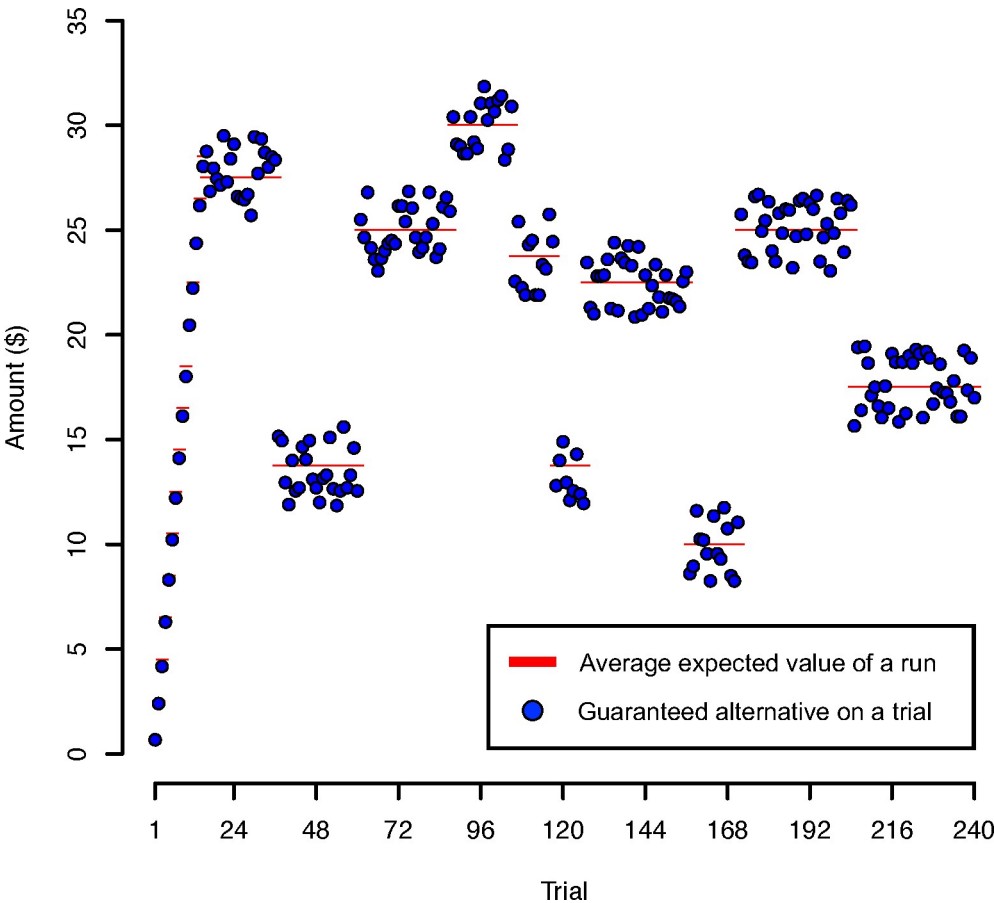

**Fig 2. Example choice set with guaranteed alternative amounts in dollars on each trial.** Risky gain values were double the guaranteed alternative values. Our novel design uses intentional temporal structure to create, change, and measure context at three timescales. We measured changes in risky decision-making behavior following outcomes on each trial (immediate timescale), large shifts in expected value following runs of trials with a shared mean expected value, or EV level (neighborhood timescale), and cumulative earnings (global timescale).

were greater than zero (5017 "responding" trials). The analyses involving SCRs do not change when including the participants removed for insufficient SCR responses (N = 15), as expected given that these participants contributed relatively few SCR trials greater than zero (a total of 490 "responding" trials across all 15 participants) to the analyses (See Supporting Information for more details). We performed post-hoc power analyses on the basis of the available sample size for the arousal-related data ("simr" package in R) [56]. The power to detect the relationship between skin conductance responses and 1) outcome was 28% (95% CI = [19.48, 37.87], α = 0.05), 2) positive shift amount was 23% (95% CI = [15.17, 32.49], α = 0.05), and 3) cumulative earnings was 94% (95% CI = [87.40, 97.77], α = 0.05). Thus, while our post-hoc power to detect relationships between skin conductance responses and global contextual effects was excellent, the levels were somewhat lower for relationships with the immediate (previous outcome) and neighborhood (positive shift) timescales, indicating greater caution in interpreting skin conductance results at those timescales.

We additionally analyzed SCRs during the decision-making phase of the task because this phase may also represent information about the context (e.g. subjective value [27]. Following the same exclusion criteria here as for SCRs following outcomes, we excluded 35 participants for failing to show consistent SCRs following more than 25% of decisions. The analyses

involving decision SCRs are best treated as exploratory for two reasons. First, more than half of the participants were excluded from this analysis (we also report results for a complementary analysis of SCRs during the decisions for all 61 participants, which do not vary from those of the more restricted group; See the Supporting Information for details). Second, relative to SCRs following outcomes, SCRs during the decision-making phase are harder to interpret as participants processed and decided between two choice options during a relatively short window of time (up to 4s).

To measure SCRs, electrodes were attached on the distal phalanges of the index and middle finger of the non-dominant hand. The SCR data were amplified and recorded with a BIOPAC Systems skin conductance module connected to an Apple computer. Data were recorded at a rate of 200 samples per second. SCR analysis was conducted using MATLAB 2018b. SCR (in microsiemens, $\mu$S) to outcomes was measured as the trough-to-peak amplitude difference in the window 0.5s to 4.5s after outcome onset. Responses below .02$\mu S$ were scored as "0". SCR was preprocessed with an FIR low-pass filter with a cutoff frequency of 25 Hz [59, 61] and mean value smoothing using a three-sample Blackman window [62]. Sixteen coefficients were used for filtering as recommended by BIOPAC (2017). Each SCR amplitude value was square-rooted to reduce skewness and normalized by each participant's maximum SCR to capture within-participant, trial-by-trial effects. Each participant's SCR on a given trial, $t$, is therefore calculated as $\frac{\sqrt{\mu S_t}}{\max\ (\sqrt{\mu S})}$.

## Results

We first replicated prior behavioral research on the influence of recent events (specifically, outcome amount) on subsequent risk-taking, and examined the presence of additional temporal contextual effects on behavior at other timescales. We then sought to extend those findings by examining arousal responses to outcomes, and the influence of those arousal responses on subsequent risk-taking.

### Risk-taking

Due to the intentional temporal structure of our choice set in which all trials within runs shared a similar EV, monetary values were correlated from one trial to the next. This collinearity across trials thus precluded the use of both trial-level and previous trial event regressors in the same regression. To address this, we took a conservative two-step approach. First, we estimated the extent to which current trial variables (e.g. risky gain$_t$, in dollars, where $t$ represents the current trial) accounted for risk-taking behavior (choice$_t$) by fitting a generalized linear binomial mixed effects model to the binary risky choice data (coded as 0 = safe choice, 1 = risky choice) using the package "lme4" [50] in R. For model 1, we included risky gain$_t$, safe$_t$, and magnitude$_t$ (which was simply the EV level of that run) because choice options on each trial were derived by the EV level plus or minus up to \$2 (for full model specifications and results, see S2 Table and the study Github repository [49]; in pseudo R code: model 1 = glmer(choice(t) ~ 0 + risky gain amount(t) + safe amount(t) + magnitude(t) + (0 + risky gain amount(t) + safe amount(t) | Subject ID), family = "binomial")). Regression estimates revealed the expected pattern of sensitivity to current trial variables. On each trial, participants were more likely to choose a given option as its value increased (risky gain$_t$ $\beta$ = 30.88(2.9), $p < 2$ x $10^{-16}$, safe$_t$ $\beta$ = -13.18(5.19), $p$ = .01), and in general, gambled relatively less when values were large (magnitude$_t$ $\beta$ = -49.51(8.9), $p$ = 2.6 x $10^{-8}$).

To examine whether recent events influenced risk-taking, in the second step of our conservative two-step approach we held the parameter estimates for the effects of current trial variables (i.e. the effects of risky gain$_t$, safe$_t$ and magnitude$_t$) constant at their previously estimated

values and added new regressors whose coefficients we then estimated. In effect this procedure allows the new regressors to attempt to account for variance not previously explained by current trial-level items (i.e. the residuals of model 1).

To execute this in R, for each trial we calculated the predicted values from model 1, prior to application of the link function, i.e. before transformation into probability space in the softmax function. These values were then included using the "offset" argument in the glmer function (see above, and see S2 Table and Analysis Files for R code). This approach is numerically identical to calculating residuals, and was used in all following linear regressions to estimate the effects of contextual regressors (see below). This two-step approach is conservative because although temporal context and current trial variables were correlated, this approach gives all shared variance to current trial variables. Estimates of the influence of contextual factors on risky decision-making can thus be considered lower bound values, due to our conservative analytical approach.

Model 2 focused on the effects of events immediately preceding subsequent choices. One recent study using a nearly identical task structure [8] found that risk-taking was significantly predicted by recent outcomes. The effect was driven by the immediate prior outcome amount above and beyond other factors such as the prior choice, mean expected value of prior choice options, or the prior outcome type (i.e. risky win, risky loss, or safe outcome). The effect of the previous outcome amount was also relatively short-lasting, best characterized as the effect of the immediately-preceding trial only. On the basis of these previous findings, we therefore added $outcome_{t-1}$ to the regression on $choice_t$, where $outcome_{t-1}$ is the dollar amount received on the previous trial (model 2 = glmer(choice(t) ~ 0 + outcome(t-1) + (1|Subject ID), family = "binomial", offset = predicted values from model 1)). We found that as $outcome_{t-1}$ increased, participants were unusually conservative on the subsequent trial, gambling less than current trial variables ($risky gain_t$, $safe_t$, $magnitude_t$) predicted (model 2; $outcome_{t-1}$ $\beta$ = -.15(.06), $p$ = .015). Th"s replicates our previous finding that risk-taking decreases following large (vs. small) outcome amounts [8], in a different set of participants using a novel choice set and is additionally consistent with two other studies [7, 11]. Because of our conservative approach which places all shared variance explained by trial-level variables and recent events into only the former, we expected a smaller estimate of the effect size than previously documented. This means that the estimated effect size should be treated as representing the lower limit of the effect size, not the true effect size itself. Nevertheless, the magnitude of this effect was such that after an outcome of +$68, participants were 3.7% less likely to take a risk than after an outcome of $0 (Fig 3A).

The results from model 2 demonstrate temporal context effects on a timescale associated with immediate, trial-by-trial feedback. We additionally sought to test whether temporal effects appeared at two other levels: the "neighborhood" level (i.e. across multiple trials), and the "global" level (i.e. across the entire task).

To test for context effects at the neighborhood level, we examined risk-taking on trials following a shift in the EV level, when the current trial suddenly differed greatly from recent context. Shifts could be positive (when values on the current trial were greater than on the previous run) or negative (when values on the current trial were smaller than on the previous run). To test whether risk-taking changed following a shift, we extended model 2 by adding regressors for positive $shift_t$ and negative $shift_t$ (corresponding to the discrepancy, in dollars, between the EV level on the current trial and the previous trial) in addition to $outcome_{t-1}$ (model 3a = glmer(choice(t) ~ 0 + outcome(t-1) + positive shift amount(t) + negative shift amount(t) + (1|Subject ID), family = "binomial", offset = predicted values from model 1)). Participants took significantly more risks following a positive shift ($\beta$ = 4.8(1), $p$ = .000001), but risk-taking did not change following a negative shift ($\beta$ = -.37(.9), $p$ = .68). Model 3a also

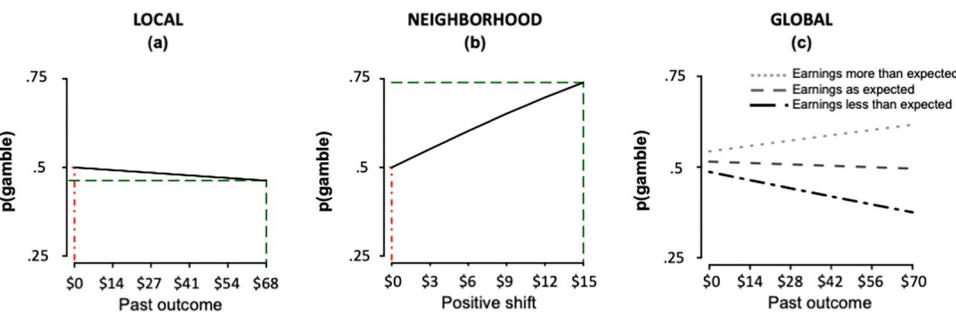

**Fig 3. Risk-taking behavior is contextually sensitive at multiple timescales.** Visualizing the effect sizes of three temporal timescales of risky decision-making. (a) Immediate timescale. Risk-taking decreases following a large outcome (green) relative to a small or negative outcome (red), assuming indifference (probability of gambling = 0.5; grey horizontal line) on the current trial. (b) Neighborhood timescale. Risk-taking increases following a large positive shift or change in mean expected value(green) relative to a trial following a small shift or negative change in mean expected value(red), assuming indifference (probability of gambling = .5; grey horizontal line) on the current trial and a past outcome of $0. (c) Global timescale interacts with immediate timescale. The effect of past outcome on risk-taking is negative when earnings are more than expected and becomes positive when cumulative earnings are more than expected (assuming indifference on the current trial).

replicated the previously identified negative effect of outcome$_{t-1}$ on risk-taking ($\beta$ = -.18(.06), $p$ = .002). These estimates indicate that, following a large shift of +$15, participants would be up to 24.2% more likely to take a risk than would otherwise have been expected (Fig 3C).

To characterize the increase in risk-taking following a positive shift, we tested how long the effect of the positive shift lasted (model 3b = glmer(choice(t) ~ 0 + outcome(t-1) + positive shift amount(t) + positive shift amount(t-1) + (1|Subject ID), family = "binomial", offset = predicted values from model 1)) and whether the effect of the positive shift interacted with the negative effect of outcome$_{t-1}$ to influence risk-taking (model 3c = glmer(choice(t) ~ 0 + outcome(t-1) + positive shift amount(t) + positive shift amount(t)*outcome(t-1) + (1|Subject ID), family = "binomial", offset = predicted values from model 3)). The effect of positive shifts was very short-lasting (model 3b; Fig 4A), with risk-taking significantly increasing only on the

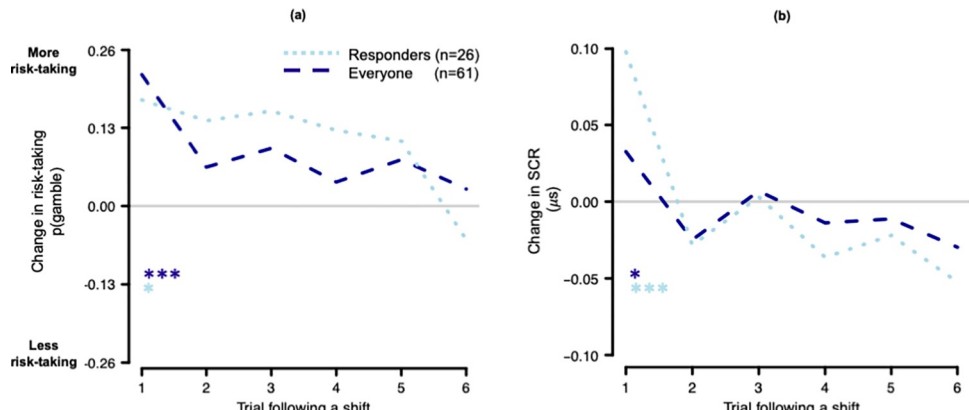

**Fig 4. Change in baseline risk-taking and physiological arousal during the decision phase following large positive shifts.** Baseline captures risk-taking and SCR ($\mu$S) following trials where a shift was not present (change in risk-taking and SCR = a large positive shift of $15 –baseline). (a) Risk-taking significantly increases on the trial immediately following a large positive shift ($15) for both responders (those participants who had sufficient numbers of trials with SCRs greater than zero) and all participants (with the exclusion of one due to equipment malfunction). (b) SCRs during the decision phase significantly increase on the trial immediately following a large positive shift for both responders and all participants, showing a similar pattern to risk-taking behavior following a large positive shift. (p < .001***, p < .05*).

trial immediately following a positive shift, and not two trials after a shift (model 3b; positive shift$_t$ $\beta$ = 4.6(1), $p$ = .000005, versus positive shift$_{t-1}$ $\beta$ = 1.5(1), $p$ = .12; this model also replicated the negative effect of outcome$_{t-1}$ $\beta$ = -.19(.06), $p$ = .001). The positive shift effect also did not appear to interact with previous outcome (model 3c), indicating that the effects of positive shift$_t$ and outcome$_{t-1}$ independently influence risk-taking (model 3c; outcome$_{t-1}$ x positive shift$_t$ $\beta$ = -7.7(5.4), $p$ = .15; outcome$_{t-1}$ $\beta$ = -.17(.06), $p$ = .005; positive shift$_t$ $\beta$ = 6.2(1.4), $p$ = .00001). See supplementary materials for additional, supporting models. Models 3a-c suggest that participants are, to some extent, tracking the recent context established by multiple trials, and that following a sudden increase in value, are unusually risk-seeking. Furthermore, these effects seem to be independent of the effects of previous outcome. That risk-taking increases on trials immediately following a positive shift (when the mean EV increases from one trial to the next) might initially appear opposite the effect of magnitude (i.e., the EV level of a given run) in model 1 where risk-taking decreases as mean EV of the current trial increases. The difference is that the magnitude effect captures overall changes in risk-taking on all trials (proximal to a shift or not) as a function of the mean EV whereas the positive shift effect is a short-lasting change in risk-taking that only occurs immediately following a large positive (and not negative) shift in expected value.

The short-lasting effect of positive shift could be due to a large change in mean expected value regardless of whether that change in value is preceded by a run of trials with a similar value. In the Supporting Information, we report an analysis that tests the effect of change in mean expected value on risk-taking in a non-structured environment (i.e., in which choice option values were not grouped in runs of shared values) and found no effects of change in mean expected value from one trial to the next on risk-taking (see Supporting Information for results and discussion). These results suggest that exposure to runs of trials with similar values are necessary to produce the effect of context at the neighborhood timescale.

Having identified independent contextual effects at the immediate and neighborhood timescales, we finally tested for the presence of temporal context effects on a third, global timescale that captured events across the entire task by examining whether cumulative earnings accounted for risk-taking behavior. Although participants were paid the outcome of one randomly selected trial, a standard protocol for avoiding "wealth effects" specifically and context effects more broadly [7, 9, 15, 22–25, 27–30, 54, 63–72], it is increasingly apparent that risky monetary decision-making is fundamentally dynamic even when the task structures are chosen to actively discourage, let alone not explicitly encourage, such behavior [8], and as observed here on the immediate and neighborhood timescales.

Because all trials featured either zero or positive monetary amounts, cumulative earnings were always non-decreasing. To capture expected earnings, we assumed linear expectations across the task (i.e. capturing the increase in earnings across the task). We represented both earnings and expectations by adding to model 3 new regressors for both cumulative earnings$_t$ (participants' cumulative earnings up until trial $t$) and linear expectations$_t$ (participants' expected earnings up until trial $t$) in addition to outcome$_{t-1}$ and positive shift$_t$ as before (model 4a = glmer(choice(t) ~ 0 + outcome(t-1) + positive shift amount(t) + cumulative earnings(t) + linear expectation(t) + (1|Subject ID), family = "binomial", offset = predicted values from model 1)). Cumulative earnings had a positive effect on risk-taking ($\beta$ = .76(.28), $p$ = .006) and linear expectations had a weak, negative effect on risk-taking ($\beta$ = -.43(.21), $p$ = .04) in addition to the effects of outcome$_{t-1}$ ($\beta$ = -.34(.08), $p$ = .000005) and positive shift$_t$ ($\beta$ = 4.8(1), $p$ = .000002). The effect of cumulative earnings is best understood by comparison to expected earnings where risk-taking increased when earnings were higher than expected and decreased when earnings were lower than expected (Fig 5). The model that accounted for both cumulative earnings and linear expectations (model 4a AIC = 13991.3) outperformed models with

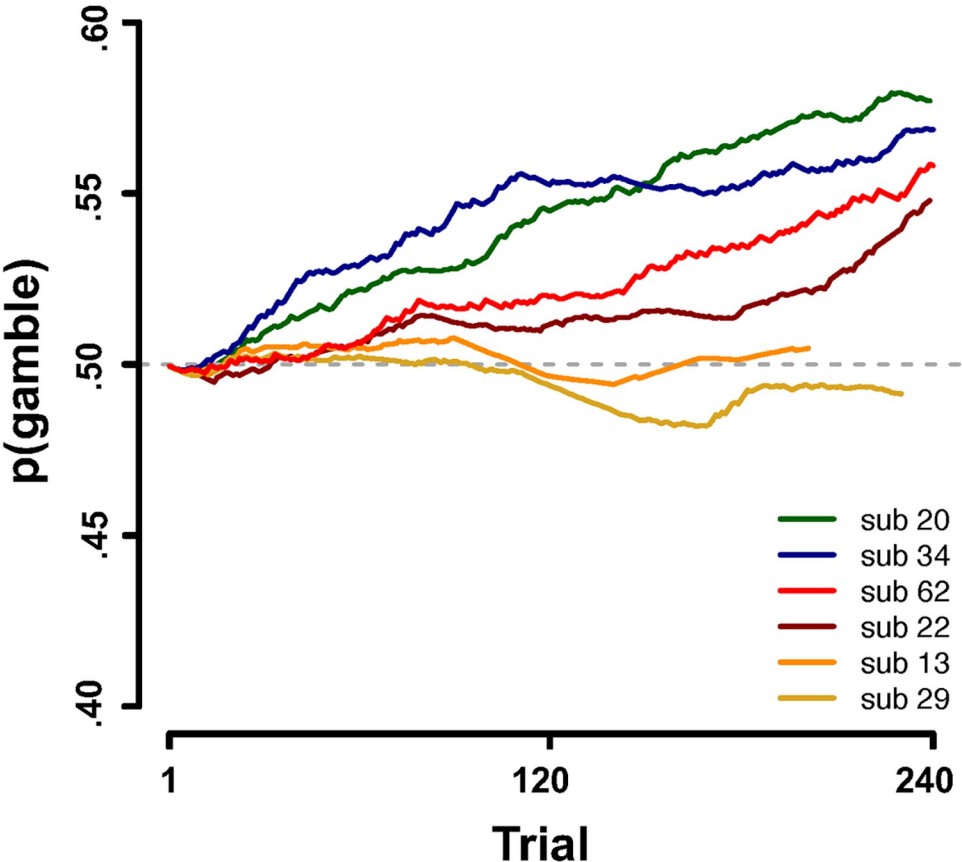

**Fig 5. The global timescale of context-dependence on risk-taking behavior.** Visualizing the combined effect of cumulative earnings and linear expectations on risk-taking for six example participants assuming indifference (probability of gambling = 0.5; grey dotted line) on the current trial, on the basis of model 4a. Each line depicts how the probability of gambling changes as a function of the positive main effect of cumulative earnings and negative main effect of linear expectations. Participants take more risks when doing better than expected (i.e. cumulative earnings are higher than linear expectations) and are less risk-taking when doing worse than expected (i.e. cumulative earnings are lower than linear expectations).

only cumulative earnings (AIC = 13993.5) or expectations (AIC = 13996.9; lower AIC is better), suggesting that participants were, to some extent, tracking their earnings relative to expectations, despite task incentives. Critically, modeling expectations and earnings as separate regressors examines, rather than assumes, their relative weighting (see Supporting Information for more discussion of this modeling approach). The relative weight of these two independent variables can capture when individuals are ahead or behind their expectations throughout the task. Note that while we assumed linear expectations across the study, it is possible that expectations are not linear. We report the results of additional analyses where we tested a piecewise linear expectation term in the Supporting Information.

The effect of large previous outcomes in reducing risk-taking seemed at odds with the effect of large cumulative earnings (relative to expectations) in increasing risk-taking. To examine these opposing influences, we tested for an interaction between these two variables in model 4b (glmer(choice(t) ~ 0 + outcome(t-1) + positive shift amount(t) + cumulative earnings(t) + cumulative earnings(t)*outcome(t-1) + linear expectation(t) + (1|Subject ID), family = "binomial", offset = predicted values from model 1)) and found a significant interaction between cumulative earnings$_t$ and outcome$_{t-1}$ ($\beta$ = 1.3(.29), $p$ = .00002) in addition to a main

effect of outcome$_{t-1}$ ($\beta$ = -.7(.11), $p$ = 3.86 x 10$^{-10}$) and positive shift$_t$ ($\beta$ = 5.3(1), $p$ = 2 x10$^{-7}$) with no main effect of cumulative earnings$_t$ ($\beta$ = .37(.29), $p$ = .2) or linear expectation$_t$ ($\beta$ = -.26 (.22), $p$ = .23). Following large outcomes, participants were less risk-taking but only when cumulative earnings were low to moderate, or less than or equal to linear expectations. As cumulative earnings increased (and became larger than linear expectations), large outcomes lead to more risk-taking (Fig 3). This interaction between cumulative earnings and previous outcome may be one explanation for the conflicting results in previous studies showing decreased risk-taking [7, 8, 11] and a mixture of increased and decreased risk-taking [9, 12–14] following large outcomes. Results from models 4a-b establish that temporal context influences risk-taking at a global timescale, but such an effect interacts with the recent events taking place on an immediate, trial-level timescale.

Our previous research on context effects [8] did not consider context effects at the neighborhood or global timescales. To check the robustness of the above behavioral findings, we performed supplementary analyses of a separate, previously collected dataset from another study [53]. In these analyses, we additionally independently replicated the effects of temporal context at three timescales. See Supporting Information for the methods, analysis approach, and results.

Taken together, our behavioral findings establish that risky monetary decision-making is dynamic at multiple timescales–immediate, reflecting the influence of the events of one trial on the next, neighborhood, reflecting the influence of sequences of trials, and global, reflecting the influence of study-level quantities like cumulative earnings (relative to expectations).

## Skin conductance responses

We next analyzed skin conductance responses (SCRs), examining the extent to which SCRs were related to risky decision-making events, and potential connections between SCRs and contextual effects on behavior. For this analysis, we include results for SCRs following outcomes (N = 46 responders) and an exploratory analysis for SCRs during the decision-phase (N = 26 responders) since both may represent information about the context [27, 29, 43, 45]. See Methods for details on SCR preprocessing and exclusion criteria).

Participants demonstrated a range of mean SCRs following wins (.03-.38 $\mu$S), losses (0-.33 $\mu$S), and safe (.03-.34 $\mu$S) outcomes. Three paired-sample t-tests revealed no significant difference in mean SCRs following wins (M = .17, SD = .08), losses (M = .167, SD = .08), and safe (M = .159, SD = .065) outcomes (wins v. losses: $t(45)$ = .34, p = .74; wins v. safe: $t(45)$ = 1.15, $p$ = .26; losses v. safe: $t(45)$ = .74, $p$ = .46), suggesting that SCRs to outcomes were not affected by the categorical type of outcome.

We next tested whether skin conductance responses to outcomes were related to the three timescales of temporal context in three linear mixed effects models with SCR$_t$ as the dependent variable. We found no significant relationship at the immediate timescale between SCR$_t$ and outcome$_t$ (model 5a = lmer(SCR following outcome(t) ~ 1 + outcome(t) + (1|Subject ID)); outcome$_t$ $\beta$ = -.013(.009), p = .16), or at the neighborhood timescale between SCR$_t$ and positive shift$_t$ (model 5b = lmer(SCR following outcome(t) ~ 1 + positive shift amount(t) + (1|Subject ID)); positive shift$_t$ $\beta$ = -.13(.1), p = .21). At the global timescale, we identified a positive effect of cumulative earnings (earnings$_t$) on SCR$_t$ but no relationship between SCR$_t$ and linear expectations$_t$ (model 5c = lmer(SCR following outcome(t) ~ 1 + cumulative earnings(t) + linear expectation(t) + (1|Subject ID)); earnings$_t$ $\beta$ = .12(.05), $p$ = .0005; linear expectations$_t$ $\beta$ = -.02 (.04), $p$ = .65). That we found no relationship between SCRs following outcomes and linear expectations (a variable that increases linearly with time) suggests that the relationship between SCRs and cumulative earnings is unlikely the result of time-on-task.

In the behavior analysis, we noted the effect of cumulative earnings on risk-taking was best characterized by an interaction with outcome (model 4b). To test for the same effect here on $SCR_t$ (an interaction between cumulative earnings and outcome to predict SCR), we regressed $SCR_t$ on an interaction term between cumulative earnings$_t$ and outcome$_t$ (model 5d = lmer (SCR following outcome(t) ~ 1 + cumulative earnings(t)*outcome(t) + (1|Subject ID))). While the main effect of cumulative earnings$_t$ remained ($\beta$ = .002(.0002), $p$ = 2 x $10^{-16}$), we found no significant interaction between cumulative earnings$_t$ and outcome$_t$ ($\beta$ = .0002(.0004), $p$ = .75). While cumulative earnings interact with outcome to account for risk-taking, the relationship between cumulative earnings and SCRs appears to be independent of immediate temporal context effects suggesting that SCRs following outcomes represent context at the global level that captures how one is doing relative to expectations. Note that because data exclusions for arousal-related data reduced the power of some of these analyses, we suggest caution especially in interpreting the null results for the immediate and neighborhood timescales. However, including data from all participants did not qualitatively change our findings (see Supporting Information for results).

**Skin conductance responses and risk-taking.**   We next tested whether physiological arousal responses to outcomes accounted for subsequent risk-taking behavior in addition to the three timescales of temporal context that we noted in the behavior analysis. If skin conductance responses were the underlying mechanism representing e.g. cumulative earnings, then we would expect that skin conductance responses would significantly predict choices subsequent choices, and outperform the cumulative earnings variable (or at least compete with it for variance).

In model 6a, we added $SCR_{t-1}$ to the regressors from model 4b including outcome$_{t-1}$, positive shift$_t$, cumulative earnings$_t$, and an interaction between outcome$_{t-1}$ and cumulative earnings$_t$; and the constant effects of current trial-level regressors fit in model 1 (glmer(choice(t) ~ 0 + outcome(t-1) + positive shift amount(t) + cumulative earnings(t)*outcome(t-1) + SCR following outcome(t-1) + (1|Subject ID), family = "binomial", offset = predicted values)). We found no additional effect of $SCR_{t-1}$ ($\beta$ = -.1(.1), $p$ = .38) on risk-taking behavior (choice$_t$) when accounting for and replicating the effects of outcome$_{t-1}$ ($\beta$ = -.75(.14), $p$ = 2.78 x $10^{-8}$), positive shift$_t$ ($\beta$ = 4.8(1.2), $p$ = .00008), cumulative earnings$_t$ ($\beta$ = .21(.09), $p$ = .017) and an interaction between outcome$_{t-1}$ and cumulative earnings$_t$ ($\beta$ = 1.15(.35), $p$ = .001).

In the behavior analysis, we observed an interaction between cumulative earnings and outcome to account for risk-taking behavior (model 4b). Given the positive relationship between cumulative earnings and SCRs (noted in model 5c), we tested whether SCRs also interacted with outcome to predict risk-taking, effectively replacing the cumulative earnings$_t$ regressor with $SCR_{t-1}$. In model 6b, we regressed choice$_t$ on an interaction between $SCR_{t-1}$ and outcome$_{t-1}$, in addition to outcome$_{t-1}$ and positive shift$_t$, omitting cumulative earnings$_t$ (model 6b = glmer(choice(t) ~ 0 + outcome(t-1) + positive shift amount(t) + SCR following outcome (t-1)*outcome(t-1) + (1|Subject ID), family = "binomial", offset = predicted values)). We found no interaction between outcome$_{t-1}$ and $SCR_{t-1}$ ($\beta$ = -.15(.42), $p$ = .73) while all other effects were consistent with our previous models (outcome$_{t-1}$ $\beta$ = -.18(.09), $p$ = .03; positive shift$_t$ $\beta$ = 4.4(1.2), $p$ = .0003; SCRs$_{t-1}$ $\beta$ = .01(.13), $p$ = .47).

The above findings establish that SCRs increase with cumulative earnings. However, while earnings interact with previous outcomes to influence risk-taking, SCRs do not directly influence risk-taking. This indicates that while SCRs and cumulative earnings are related, they are not exchangeable, and suggests that they may carry different information about context and/or have separable roles in risky decision-making. Perhaps most importantly, they also suggest that skin conductance responses may result from cumulative earnings representations, and not causally underlie them.

**Skin conductance responses during the decision phase.** For this exploratory analysis, we tested whether SCRs during the decision phase were related to temporal context and whether potential relationships between SCRs and temporal context accounted for risk-taking behavior. The decision phase encompassed the entire time from when decision options were initially presented, through the forced viewing period (2s) and the response window (up to 2s), up until the moment a decision was entered by a button press (Fig 1). While SCRs during the decision-phase may relate to context in risky decision-making, these results are especially difficult to interpret given the number of cognitive and affective events, including evaluating each option separately, comparing them, possibly integrating context into valuation and action judgements, and then executing an action. We thus treated the following analyses as exploratory, due to the impossibility of fully dissociating the various processes occurring during the decision process.

Participants demonstrated a range of mean SCRs when taking risks (.03-.36 $\mu$S) and when rejecting risks for the guaranteed alternative (.05-.33 $\mu$S). We examined whether SCRs during the decision phase were related to choices in two ways. First, a paired-samples Wilcoxon signed-rank test revealed no overall significant difference in mean SCRs by decision type (gamble M = .135, SD = .22; guaranteed alternative M = .146 SD = .22; $V$ = 173, $p$ = .96). Second, a linear mixed effects model regressing SCRs on decisions (coded as 1 = risky choice, -1 = safe choice) revealed no significant change in SCRs as a function of risk-taking (choice $\beta$ = .0008(.003), $p$ = .78; model 7 = lmer(SCR during decision phase(t) ~ 1 + choice(t) + (1|Subject ID))). These results suggest that SCRs during the decision phase were not straightforwardly related to accepting or rejecting the gamble.

We next tested whether SCRs during the decision phase varied as function of the three timescales of temporal context by regressing $SCR_t$ on $outcome_{t-1}$, positive $shift_t$, and cumulative $earnings_t$ (accounting for linear $expectations_t$; model 8a = lmer(SCR during decision phase(t) ~ 1 + outcome(t-1) + positive shift amount(t) + cumulative earnings(t) + linear expectation(t) + (1|Subject ID))). We identified a weak effect of $outcome_{t-1}$ ($\beta$ = -.02(.01), $p$ = .05), a main effect of positive $shift_t$ ($\beta$ = .41(.13), $p$ = .002), and no effect of cumulative $earnings_t$ ($\beta$ = -.04(.06), $p$ = .45) or linear $expectations_t$ ($\beta$ = .07(.05), $p$ = .14) on decision $SCR_t$. These results indicate that SCRs during the decision phase increase following large positive shifts, varying at the neighborhood level of temporal context, but not at the immediate (i.e. $outcome_{t-1}$) or global (e.g. cumulative $earnings_t$) levels of temporal context.

To characterize the effect of positive shift on SCRs during the decision phase, we examined how long this effect lasted (model 8b = lmer(SCR during decision phase(t) ~ 1 + positive shift amount(t) + positive shift amount(t-1) + (1|Subject ID))). In Model 8b, we regressed $SCR_t$ on to positive $shift_t$ and positive $shift_{t-1}$. The positive shift effect on SCRs during the decision phase was short-lasting, dropping off after the trial following the shift (positive $shift_t$ $\beta$ = .4 (.13), $p$ = .002; positive $shift_{t-1}$ $\beta$ = -.17(.13), $p$ = .19; Fig 4B). This positive, short-lasting effect of positive shift on SCRs during the decision phase resembles the short-lasting effect of positive shift on risk-taking behavior (i.e. increased risk-taking immediately following a positive shift).

That SCRs during the decision phase were related to positive shifts suggests that heightened arousal following a positive shift could be a reasonable underlying mechanism for the behavioral positive shift effect on choices. To test this possibility, we examined whether SCRs during the decision phase accounted for risk-taking behavior in addition to the three levels of temporal context previously identified by regressing $choice_t$ on $SCR_t$, $outcome_{t-1}$, positive $shift_t$, cumulative $earnings_t$, and an interaction between $outcome_{t-1}$ and cumulative $earnings_t$ (model 8c = glmer(choice(t) ~ 0 + outcome(t-1) + positive shift amount(t) + cumulative earnings(t) *outcome(t-1) + SCR during decision phase(t) + (1|Subject ID), family = "binomial", offset = predicted values)). If the behavioral positive shift effect was caused by arousal

responses during those shifts, we might expect SCRs during the decision period to better account for the positive shift effect on choices. However, while the effect of $outcome_{t-1}$, positive $shift_t$, and the interaction between $outcome_{t-1}$ and cumulative $earnings_t$ remained ($outcome_{t-1}$ $\beta$ = -.78(.18), $p$ = .00002; positive $shift_t$ $\beta$ = 4.5(1.6), $p$ = .006; $outcome_{t-1}$ x cumulative $earnings_t$ $\beta$ = 1.35(.46), $p$ = .003), there was no significant effect of SCR on risk-taking ($SCRs_{t-1}$ $\beta$ = -.04 (.1), $p$ = .77). This indicates that while the positive shift effect on behavior may have its roots in e.g. surprise or novelty responses (see Discussion), physiological arousal responses are likely the result and not the cause, or are simply occurring in parallel.

To complement our correlational analysis of the relationship between temporal context, risky decision-making and skin conductance responses, we reanalyzed risky monetary choices from a previous study [53] following the administration of propranolol and a placebo, as propranolol manipulates the neurohormonal system underlying arousal responses like skin conductance. As these data differed in critical ways from the study design here, we report and discuss the results of the reanalysis in the Supporting Information. In brief, consistent with the above analyses showing that skin conductance responses correlated with contextual variables but did not appear to drive choice behavior directly, our reanalysis suggests no effect of propranolol in attenuating context dependency in risky decision-making. See Supporting Information for more details.

**Results summary.** We examined the extent to which risk-taking and physiological arousal were related to three levels of temporal context and the potential relationship between arousal and context-dependence in risky monetary decision-making. We measured SCRs during a novel risky monetary decision-making task with a unique temporal structure. At the immediate timescale, risk-taking decreased following large previous outcomes. At the neighborhood timescale, risk-taking increased following large positive shifts between runs. At the global timescale, cumulative earnings interacted with previous outcomes to decrease risk-taking following large previous outcomes only when cumulative earnings were lower than or as expected. These findings clearly establish multiple timescales of temporal context effects in risk-taking.

We found less consistent patterns when examining physiological arousal and temporal context in risk-taking. SCRs following outcomes increased as cumulative earnings increased and SCRs during the decision-phase increased following a positive shift. However, we found no evidence that SCRs interacted with or replaced temporal context variables to influence risk-taking directly. That arousal was related to two of the three timescales of temporal context examined suggests that arousal may be related to temporal context effects in risk. While our evidence suggests that arousal is the consequence and not the cause of context dependency in risky decision-making (see Supporting Information for more discussion), arousal may fulfill other roles including self-signaling, altering other processes like attention or memory, or some other role not yet understood.

## Discussion

Here, we demonstrated that temporal context simultaneously influences risky monetary decision-making on at least three timescales: immediate, neighborhood and global.

At the immediate level of temporal context, risk-taking decreased following large past outcomes, consistent with previous studies [7, 8, 11]. This pattern of behavior is consistent with behavior in environments where relying on previous outcomes is informative to subsequent decisions (e.g. where outcome probabilities are unknown or changing) [76]. However, relying on previous outcomes in a setting where probabilities are explicit and unchanging, and where outcomes do not causally influence subsequent outcomes, does not appear advantageous and

may lead to lower payoffs. Why we observe context dependence at the immediate timescale is as yet unclear. Some possibilities include explicit heuristics [73] or policies [74] that individuals might use implicitly or explicitly in the pursuit of goals [75].

At the neighborhood level, risk-taking changed briefly following large differences between the mean expected value on the current trial and the average mean expected value across several previous trials. These results indicate that individuals not only track values encountered in the recent past but that large changes in context lead to brief changes in decision-making behavior. These results are consistent with recent studies on context in other decision-making tasks demonstrating that behavior changes following shifts in context [20, 21, 23]. While the underlying mechanism and precise computations that support tracking the current context and adjusting behavior to changes in context is unclear, one compelling possibility is that individuals hold value-based expectations based on recent history which are then violated by large changes. Because existing evidence of the neighborhood timescale comes from previous studies where blocks of similar trials occur on the order of 100–300 trials (in comparison to current study with 9–36 trials in a run), it is possible that the short-lasting effect of positive shift in the current study is explained by a mechanism or computation that may occur on a shorter timescale, such as surprise, novelty, or reward prediction error. While the effect of positive shift extends only to the trial immediately following a shift, supplementary reanalyses of previously-collected data suggest that positive shift effects are only observed when taking into account multiple preceding trials at a time. However, many questions remain. For example, it is unclear why we observed changes in risk-taking following positive, but not negative shifts and whether this was related to the nature of a gain-only task (e.g. positive shifts may induce novelty or surprise because the values encountered following a positive shift are likely more novel/infrequent, whereas negative shifts lead to previously encountered values). It is also unclear whether behavioral changes at the neighborhood timescale are supported by additional cognitive processes such as attention. To examine questions like these and thereby understand the potential mechanism and computation supporting contextual effects at the neighborhood timescale, future research should examine how contextual effects in risk might change across structured and non-structured environments, and better establish the necessary conditions for shift effects to emerge.

At the global timescale, cumulative earnings interacted with past outcomes to influence risk-taking such that risk-taking decreased following large positive outcomes when cumulative earnings were less than expected. Such an interaction suggests that the context-dependent effects at the global timescale related to an individual's overall task performance relative to expectations may determine how an individual responds to events at the immediate timescale (e.g. trial-by-trial outcomes). By comparing earnings relative to inferred expectations, we noted changes in risk-taking behavior as a function of departures from expectations (and not changes in expectations themselves) suggesting that expectations may not adjust on the timescale in our task but require much longer timescales for adaptation consistent with [20]. It is important to note that expectations were inferred in the current study, and not directly measured. While linearly increasing expectations were a reasonable assumption in the current study, expectations must undoubtedly change under some conditions, subject to the precise computational form they take. Future research should consider more direct tests, manipulations, and measures of expectations themselves. Nevertheless, our results suggest that capturing context effects at the global timescale requires both earnings and their relationship to expectations.

In our study, cumulative earnings were normatively irrelevant. They did not explicitly influence outcomes or the final payment in the current task, but their effects were nonetheless observed, suggesting that the dominant frameworks for risky decision-making may have

overlooked fundamental aspects of value computation and decision-making over time. Recent research has demonstrated that individuals implicitly track cumulative earnings [11], a finding which we extend to settings in which doing so appears irrelevant, and possibly disadvantageous, to payoff (an observation that additionally applies to all context effects observed in the current study). We assumed that cumulative earnings were compared to expectations via a straightforward linear additive mechanism. While decision-makers are known to dynamically track current resources compared to a threshold or expectation in other domains of decision-making (e.g. foraging) [76], which we discuss more below), the motivations for and precise functional form of such tracking in risky monetary decision-making is unclear. It is additionally possible that the effect of earnings relative to expectations is nonlinear. In particular, some evidence [12–14] suggests that any departure of earnings from expectations might lead to more risk-taking. Fitting such a form in future research will require advanced nonlinear approaches not used here as well as larger datasets with more trials and explicit measures of expectations and earnings that are dissociated from time. What is clear is that the study of risky monetary decision-making has overlooked a critical component of choice behavior.

The effects of cumulative earnings are additionally notable because the dataset here (like many others) [7, 23, 27–29] used a payout structure designed to avoid such global context effects. It thus seems likely that the implicit or explicit tracking of earnings is not in response to task demands but is instead deeply integrated in valuation, perhaps as a kind of continual resource tracking, a potential mechanism that we discuss below. While concepts like the "reference point" and "status quo" have been significant parts of major theories on risky decision-making [26, 77] their role has also been underappreciated empirically. Clearly, risky monetary decision-making is fundamentally contextually-dependent beyond the previously established immediate timescale [8], changing as a function of multiple timescales.

The models used to analyze behavior in the current study were all relatively simple linear models (i.e. regressions), but it is possible that a more sophisticated model would fit the data better. Recent promising advances in model evaluation, comparison, and prediction have enabled researchers to test and compare the performance of many different and complex models [78–80], though findings are not yet consistent across these studies. Future research may well seek to leverage insights from this emerging body of work to incorporate temporal context variables in these models and their comparisons.

It may be tempting to explain this dynamism as resulting from 'learning,' but we feel this explanation to be unlikely. As discussed in detail elsewhere [8], we believe this for two main reasons: 1) the task itself is simple and fully explicit, with all probabilities and values communicated and instructed in detail, and comprehension of those details assessed with basic quizzes, and 2) participants do not report 'learning'-like strategies. Thus, there is nothing to learn and no reported evidence that participants are learning anything in a classic reinforcement-learning sense. Learning is not, however, the only explanation possible for dynamic behavior like that observed here, in which participants are clearly and undoubtedly responding dynamically to events in the task as they evaluate options and make choices.

Given the established central role of physiological arousal in risky decision-making, as well as the temporal characteristics of arousal, we tested physiological arousal as a potential underlying mechanism of temporal context in risk-taking. We demonstrated that skin conductance responses were associated with the neighborhood and global timescales of temporal context. These results extend the involvement of physiological arousal to context dependence in risky monetary decision-making. That positive shifts (neighborhood timescale) were associated with brief increases in skin conductance responses and risk-taking is consistent with a previous study demonstrating increases in pupil diameter and learning rates following environmental changes [81]. This convergent evidence suggests that arousal may prepare an individual to

integrate new incoming information. We urge caution in overinterpretation of these results, however, as statistical power was somewhat reduced for some analyses due to the low sensitivity in responses to task events in the current dataset, as well as the possibility that SCRs may habituate over time. Additionally, the long-lasting nature of SCRs can make their interpretation difficult because of responses potentially overlapping in time. However, skin conductance responses are but one measure of arousal, let alone affect. Future studies could examine relationships between the three timescales of context in risky decision-making settings and additional different physiological measures including pupillometry and neuroimaging, or other affective measures such as reported valence and arousal [82].

Multiple timescales of context dependence in risk-taking raise the possibility of multiple potential underlying mechanisms. For example, the mechanism supporting trial-by-trial changes in risk-taking may involve reward processing [83] whereas brief changes in behavior as a result of positive shifts in value may be the result of a simpler novelty or Pavlovian-like response [84]. The global timescale of earnings and expectations might result from a combination of mechanisms including reward processing [11, 83], reward rate information [85], and expectation tracking [11]. If different mechanisms support separable aspects of context dependence in risky financial decision-making, then future research might expect experimental manipulations or individual differences to relate to very specific aspects of context dependence. For example, if the effect of previous outcomes on risk-taking is a consciously accessible process, then this local effect may be mitigated by behavioral change strategies, such as shifting perspective [18, 29, 86–88] while other effects, like positive shifts in value at the neighborhood level, may not if they are sufficiently automatic in origin.

Some recent models of decision-making have integrated contextual features like feedback [89–91], and indeed early observations of behavioral tendencies like the gambler's fallacy (e.g., [92]) or the hot hand effect [93] clearly illustrated the importance of context in decision-making. Importantly, in many of these cases, choice objects were complex with unknown, varying, and/or multiple probabilities, raising the possibility that factors related to processing complex choice options (like attention, memory, numeracy, etc.) may be critical. Here, by simplifying the decision space (only zero or positive values were encountered; all probabilities were either 0.5 or 1; all probabilities and values were explicitly communicated; only two choice options were available at any given time), our findings extend and complement this previous work by illustrating not one but multiple levels of temporal context dependency in even the simplest of risky decision-making tasks.

There are conceptual parallels between our analysis of context dependence and the study of context in other domains. For example, recent research in memory has examined event boundaries, showing significant downstream consequences, for example, for the structure of representations, binding between mnemonic items, and thus the influence of those separate events on subsequent actions and behaviors [94–97]. The understanding of neighborhood timescale context effects in decision-making may in particular benefit from integration of event boundary research within the domain of memory. Additional insights may also be gained from the reinforcement learning literature, which has routinely leveraged concepts of surprise [98, 99] or states [100], meta-processes that arguably operate exclusively at the level of context (e.g., Pearce-Hall) [101]. The idea of tracking resource attainment at the global level and comparing it to expectations also has clear parallels to the foraging literature [76], a central feature of which is often the comparison of current earnings to a calculated expected reward rate within an environment [102], not unlike the earnings and expectations terms used here. Comparing current earnings to expectations also suggests an underappreciated role for goals in risky decision-making instead of, or in addition to, the assumed goal of reward maximization. Goals are important in value-based decision-making [75, 103] but the context effects that

we identified here (i.e. relying on recent events that have no causal influence on subsequent rewards) would suggest that individuals are not strictly maximizing potential payoffs and may instead be seeking to attain a particular rate of reward accumulation. This explanation is consistent with previous work showing increases in risk-taking when cumulative reward falls below a threshold and as foraging opportunities decrease [104]. Understanding individuals' goals and how such goals may evolve over time will be critical to understanding why we see context-dependence in risk. There are many other examples of contextual influence in human cognition (divisive normalization [20], serial dependence [105]), but a full review of the topic is beyond the scope of the present article. It is beyond doubt, however, that these many literatures converge to argue that context matters, an assertion that we have developed and refined here in the domain of simple risky decisions.

While these findings, and other recent research, establish the critical importance of temporal context to risky decision-making, many questions remain. For example, future research could examine the possible non-linearity of these contextual influences, the boundaries between the different timescales of contextual influence, the role of time itself (e.g. seconds, minutes, etc.) versus events (e.g. the most recent outcome), the possibility of temporal context effects on a scale of hours, days, weeks, or even months or longer, and so on.

Together, these findings suggest that risky monetary decision-making, often thought of as reflecting static preferences, is instead simultaneously contextually-dependent at multiple timescales, and that the role of physiological arousal in risky decision-making extends to temporal context. Remarkably, these contextual influences appear in fully-instructed laboratory settings with incentives actively designed to counter contextual effects. Real-world financial risk-taking may thus be even more influenced by temporal context because of the rich experiences that individuals bring with them. It therefore seems possible that such contextual effects will only be more relevant beyond the lab. Developing the science of risk-taking will clearly require examining and quantifying the underappreciated roles of context in risky choices.

## Supporting information

**S1 Text. Supplemental analyses, results, and discussion.**
(DOCX)

**S1 Table. Summary of task design in main text and supplement.**
(PDF)

**S2 Table. Generalized linear modeling results.**
(PDF)

## Author Contributions

**Conceptualization:** Hayley R. Brooks, Peter Sokol-Hessner.

**Formal analysis:** Hayley R. Brooks.

**Methodology:** Hayley R. Brooks.

**Project administration:** Hayley R. Brooks.

**Resources:** Peter Sokol-Hessner.

**Supervision:** Peter Sokol-Hessner.

**Visualization:** Hayley R. Brooks.

**Writing – original draft:** Hayley R. Brooks.

**Writing – review & editing:** Hayley R. Brooks, Peter Sokol-Hessner.

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
