## [Decision Letter · Decision Letter 0]

18 Sep 2023

PONE-D-23-21525Multiple timescales of temporal context in risky choice: Behavioral identification and relationships to physiological arousalPLOS ONE

Dear Dr. Sokol-Hessner,

Thank you for submitting your manuscript to PLOS ONE. After careful consideration, we feel that it has merit but does not fully meet PLOS ONE’s publication criteria as it currently stands. Therefore, we invite you to submit a revised version of the manuscript that addresses the points raised during the review process.

We look forward to receiving your revised manuscript.

Kind regards,

Rei Akaishi

Academic Editor

PLOS ONE

Additional Editor Comments:

The research conducted by Brooks and Sokol-Hessner presents intriguing experiments that demonstrate the independent influences of reward history at multiple time scales on risk-taking behavior. The manuscript is generally well-crafted; however, the models posited for underlying mechanisms lack specificity due to the absence of detailed computational models. While this ambiguity leaves room for robust discussion concerning potential mechanisms, the authors have commendably summarized relevant, particularly recent, literature. I have two primary suggestions for additional scholarly works that could enrich the manuscript's discussion section.

Firstly, the work of Kolling et al. provides precedent for the modulation of risk-taking decisions based on contextual variables. Specifically, they investigated the influence of time pressure on the attainment of a specified level of cumulative reward. I recommend that the authors consider comparing their findings with those of Kolling et al. to more comprehensively address the contextual effects on risk-taking decisions.

Reference:

Kolling, N., Wittmann, M., & Rushworth, M. F. (2014). Multiple neural mechanisms of decision making and their competition under changing risk pressure. Neuron, 81(5), 1190-1202.

Secondly, the relationship between arousal and contextual shifts has previously been explored by Joshua Gold and Matt Nassar. They assessed the impact of sudden environmental changes on arousal systems and subsequent behavioral adjustments. A comparison between these findings and the current study could provide valuable insights.

Reference:

Nassar, M., Rumsey, K., Wilson, R. et al. Rational regulation of learning dynamics by pupil-linked arousal systems. Nat Neurosci 15, 1040–1046 (2012).

Minor Point:

To enhance clarity concerning the regression analyses, I concur with Reviewer 2 in suggesting that the authors delineate all the equations related to the regression models, either in the Methods or Results section. Providing tables containing estimated coefficients, statistical significance, and criteria for fitting performance (e.g., AIC, BIC, LOOC) would be beneficial for the reader.

Reviewers' comments:

Reviewer's Responses to Questions

**Comments to the Author**

1. Is the manuscript technically sound, and do the data support the conclusions?

Reviewer #1: Yes

Reviewer #2: Partly

2. Has the statistical analysis been performed appropriately and rigorously? 

Reviewer #1: Yes

Reviewer #2: Yes

3. Have the authors made all data underlying the findings in their manuscript fully available?

Reviewer #1: Yes

Reviewer #2: Yes

4. Is the manuscript presented in an intelligible fashion and written in standard English?

Reviewer #1: Yes

Reviewer #2: Yes

5. Review Comments to the Author

Reviewer #1: This paper presents an analysis of temporal context in risky decision making, including a measure of physiolgoical arousal (skin conductance response). The key findings are that large positive outcomes increased risk-taking at short and intermediate timescales, and decreased risk-taking at long timescales (but only when cumulative earnings were sufficiently low relative to expectations). The relationship between skin conductance and risk-taking at different timescales showed several effects but lacked a clean correspondence with the behavioral findings.

Overall, I thought this was a very comprehensively analyzed dataset. The paper is clearly written and well organized. I think these findings will be very valuable to any researcher working in this area. I didn't see any major issues in the manuscript. Of course, the theoretical intepretation of these results, given the somewhat inconsisent nature of the skin conductance data, remains to be seen, but I think this kind of paper serves as a good challenge for theorists.

Minor:

p. 10: "built in into" -> "built into"

p. 27: "may have it roots" -> "may have its roots"

Reviewer #2: This manuscript by Roper and Sokol-Hessner examines how risky decision-making in human subjects depends on contextual factors, specifically past outcomes at immediate, short-term, and long-term timescales. Using a standard risk decision-task with fluctuating temporal structure (primarily runs of trials at given EVs, switching in unsignaled blocks), the authors show via regression analyses that risk taking depended on context at different timescales: immediate (past trial outcome), local neighborhood (post-run shift effects), and global (cumulative potential earnings compared to expectation). Skin conductance responses were related to the local and global timescales, but were not directly related to changes in risky choice behavior. This is an interesting set of findings that relate to a growing literature on contextual effects in economic choice behavior. The authors provide a nice extension to some of their previous work by examining temporal context effects at different timescales. The paper is on the while relatively clearly written, but the authors should clarify some issues of analysis and interpretation.

Major points

(1) Multiple timescales versus a time-varying influence

One of the most interesting aspects of the results is the finding of behavioral temporal context effects at different time scales. However, the authors could do a better job justifying their particular choice of analyses to examine short/medium/long timescales. For example, the influence of past outcomes at varying timescales have been examined in many different scenarios as a weighted (e.g. exponential kernel) sum of past trial rewards (Khaw et al for reward adaptation in behavior, Sugrue et al for matching law neural coding, etc.). Did the authors try a similar approach to determine the effect of past trials (e.g. a lagged regression)?

The neighborhood effect the authors find is interesting, but it’s not clear to me that it’s strictly a local/medium timescale effect. The authors show a change in risk-taking following a shift in EV between runs, but only on the first trial following the shift (and only for positive shifts). This seems to argue against a strict intermediate-timescale effect, which should persist for trials (as an intermediate-defined context - like running average - changes incrementally with additional trials after the shift). Rather than a temporal effect, this seems like an effect more consistent with a violation of expectation or a novelty effect (something the authors touch on in the Discussion). Additionally, can the authors discuss why they thing the effect is evident for positive but not negative shifts?

(2) Clarity about regression analysis.

The paper largely relies on regression analyses to quantify context effects, and there are a few clarifications the authors could provide.

I am concerned there may be an issue with multicollinearity in Model 1 between risky gain, safe gain and magnitude. Since the task structure changes all three variables together across different trials runs (all change together with run-specific EV level), across all trials these variables may significantly covary - making parameter estimation uncertain. Any parameter estimate problems here would affect subsequent analyses that fix these regression weights while fitting others. Did the authors test for multicollinearity (e.g. by examining VIFs) in this initial regression?

For the previous trial analyses, it seems that outcome at t-1 is potentially confound with win versus loss; I believed this was examined in the authors’ previous paper but not examined here - at least please clarify that this was the assumption, and justified by previous findings.

Additionally, I am concerned that the previous trial outcome effects (Model 2) and the neighborhood shift effects (e.g. Model 3a) reflect the same phenomenon. Model 2 shows that low past trial t-1 outcome increases risk taking. Model 3a shows that positive shifts increase risk taking, using a regressor that corresponds to the difference in EV between current trial and past trial. This regressor on shift trials covaries with the EV of the previous trial, which is correlated with the previous trial outcome. I realize the previous trial outcome is included as a covariate, but it would help if the authors examined the collinearity between shift and outcome regressors.

(3) Global context analysis.

The authors could provide a little more explanation about what the global context analyses are meant to capture exactly. The results are framed as the overall effect of a difference between cumulative earnings experienced and a linear expectation, and at first glance the relevance of these constructs to subjects is unclear. If the subjects are explicitly told that they would be rewarded for a single trial, why would they track the cumulative earnings (rather than, for example, the average)? It’s also unclear why subjects would have a linear, constantly increasing expectation; testing the piecewise linear expectation was a good addition, but still has the issue of an increasing expectation not matching the single rewarded trial structure.

In addition, it seems that both these quantities - and in particular, the linear expectation - would be confounded by a simple measure of time in the experimental session. Is there a similar effect if trial number is included in the regression?

Minor points

(1) It would greatly help clarity if the authors provided equations for their different regression models (preferably in the Methods, though SI would be OK). I think grouping them together in one place would help the reader see the differences and similarities between the different model versions.

(2) Small clarification: were the L/R screen position of safe/risky gambles counterbalanced or randomized across trials?

(3) line 49: not clear what “risk-taking against disincentives” means; perhaps “despite incentives” might be clearer?

6. PLOS authors have the option to publish the peer review history of their article (what does this mean?). If published, this will include your full peer review and any attached files.

Reviewer #1: No

Reviewer #2: No

---

## [Author Response · Author response to Decision Letter 0]

22 Nov 2023

We attached a Response to Reviewers document that comprehensively addresses each question or concern of the Editor and reviewers, including formatting, tables, and figures where appropriate. We deeply appreciate the engagement and constructive criticism of the Editor and reviewers, and believe the manuscript is greatly improved as a result. Please see the Response to Reviewers document for the full point-by-point response. Thank you!

---

## [Decision Letter · Decision Letter 1]

18 Dec 2023

Multiple timescales of temporal context in risky choice: Behavioral identification and relationships to physiological arousal

PONE-D-23-21525R1

Dear Dr. Sokol-Hessner,

We’re pleased to inform you that your manuscript has been judged scientifically suitable for publication and will be formally accepted for publication once it meets all outstanding technical requirements.

Kind regards,

Rei Akaishi

Academic Editor

PLOS ONE

Additional Editor Comments (optional):

Reviewers' comments:

Reviewer's Responses to Questions

**Comments to the Author**

1. If the authors have adequately addressed your comments raised in a previous round of review and you feel that this manuscript is now acceptable for publication, you may indicate that here to bypass the “Comments to the Author” section, enter your conflict of interest statement in the “Confidential to Editor” section, and submit your "Accept" recommendation.

Reviewer #1: All comments have been addressed

Reviewer #2: All comments have been addressed

2. Is the manuscript technically sound, and do the data support the conclusions?

Reviewer #1: Yes

Reviewer #2: Yes

3. Has the statistical analysis been performed appropriately and rigorously? 

Reviewer #1: Yes

Reviewer #2: Yes

4. Have the authors made all data underlying the findings in their manuscript fully available?

Reviewer #1: Yes

Reviewer #2: Yes

5. Is the manuscript presented in an intelligible fashion and written in standard English?

Reviewer #1: Yes

Reviewer #2: Yes

6. Review Comments to the Author

Reviewer #1: (No Response)

Reviewer #2: This is a revised manuscript by Brooks and Sokol-Hessner examining how risky decision-making in human subjects depends on contextual factors, specifically past outcomes at immediate, short-term, and long-term timescales. The authors have done a nice job of addressing my previous concerns, and I appreciate their additional analyses and revisions to the text.

7. PLOS authors have the option to publish the peer review history of their article (what does this mean?). If published, this will include your full peer review and any attached files.

Reviewer #1: No

Reviewer #2: No

---

## [Editor Report · Acceptance letter]

9 Jan 2024

PONE-D-23-21525R1 

PLOS ONE

Dear Dr. Sokol-Hessner, 

I'm pleased to inform you that your manuscript has been deemed suitable for publication in PLOS ONE. Congratulations! Your manuscript is now being handed over to our production team.

Kind regards, 

on behalf of

Dr. Rei Akaishi 

Academic Editor

PLOS ONE